# Structural and Functional Analysis of BBA03, *Borrelia burgdorferi* Competitive Advantage Promoting Outer Surface Lipoprotein

**DOI:** 10.3390/pathogens9100826

**Published:** 2020-10-09

**Authors:** Jēkabs Fridmanis, Raitis Bobrovs, Kalvis Brangulis, Kaspars Tārs, Kristaps Jaudzems

**Affiliations:** 1Department of Physical Organic Chemistry, Latvian Institute of Organic Synthesis, Aizkraukles 21, LV-1006 Riga, Latvia; fridmanis.jekabs@osi.lv (J.F.); raitis.bobrovs@osi.lv (R.B.); 2Latvian Biomedical Research and Study Centre, Ratsupites 1, LV-1067 Riga, Latvia; kalvis@biomed.lu.lv (K.B.); kaspars@biomed.lu.lv (K.T.); 3Department of Human Physiology and Biochemistry, Riga Stradins University, Dzirciema 16, LV-1007 Riga, Latvia; 4Department of Molecular Biology, University of Latvia, Jelgavas 1, LV-1004 Riga, Latvia; 5Department of Organic Chemistry, University of Latvia, Jelgavas 1, LV-1004 Riga, Latvia

**Keywords:** BBA03, Pfam54, competitive advantage, lp54, solution NMR structure

## Abstract

BBA03 is a *Borrelia burgdorferi* outer surface lipoprotein encoded on one of the most conserved plasmids in *Borrelia* genome, linear plasmid 54 (lp54). Although many of its genes have been identified as contributing or essential for spirochete fitness in vivo, the majority of the proteins encoded on this plasmid have no known function and lack homologs in other organisms. In this paper, we report the solution NMR structure of the *B. burgdorferi* outer surface lipoprotein BBA03, which is known to provide a competitive advantage to the bacteria during the transmission from tick vector to mammalian host. BBA03 shows structural homology to other outer surface lipoproteins reflecting their genetic and evolutionary relatedness. Analysis of the structure reveals a pore in BBA03, which could potentially bind lipids.

## 1. Introduction

Lyme disease is the most widespread arthropod-transmitted disease in North America and Europe [1]. It is caused by *Borrelia burgdorferi sensu lato* complex spirochetes, which are transferred to a mammalian host through a bite of an infected tick [2]. If the infection is diagnosed at an early stage, 14 day oral antibiotic treatment is sufficient to cure Lyme disease. If not treated or treated inadequately, it can spread to major organs within the host, resulting in severe skin, joint, or neurological problems [3,4].

One of the most highly conserved plasmids in the segmented *Borrelia burgdorferi* genome is lp54 [5]. Several of its encoded proteins such as OspA that is required for persistence within the tick and DbpA/B decorin binding proteins that are important for dissemination in the mammal have been identified as essential for spirochete survival in vivo [6,7]. Others such as BBA52, BBA64, and BBA03 play a significant role in the transmission from tick to mammals, but their specific function has not been identified thus far [8,9,10]. The expression of BBA03 increases six times during the tick blood meal and decreases only two-fold from its initial value one week after the infection. Incubation of whole bacteria in presence of proteinase K does not result in degradation of BBA03, whereas immunofluorescence data indicate that it is at least partially surface exposed [8]. BBA03 deletion mutant, although fully competent by itself, cannot infect mice when transmitted by ticks that were simultaneously coinfected with wild-type *B. burgdorferi*. The competitive advantage of BBA03 is absent in case the bacteria are inoculated with a syringe. Lack of BBA03 also does not affect migration of *Borrelia* from tick midgut to salivary glands. This implies a role for BBA03 in transition from tick salivary glands to the mammalian host, suggesting it as a potential target for vaccine development. However, mice immunized with BBA03 showed only minor protection against *B. burgdorferi* infection by a tick bite [8].

Here, we report the solution NMR structure of BBA03 from *B. burgdorferi* B31 strain. Structural analysis and the known function of its identified structural homologs suggest that BBA03 may function as a lipid binding protein.

## 2. Results

### 2.1. BBA03 Structure Determination

Uniformly ^13^C,^15^N-labeled BBA03_23–169_ protein was expressed in *Escherichia coli* BL21(DE)3 strain and purified to apparent homogeneity as described in Materials and Methods Section 4.1. Its monomeric state was confirmed by NMR ^15^N spin-lattice (T_1_) and spin-spin (T_2_) relaxation experiments (Appendix A). Chemical shift assignments for the backbone and the ^13^C_β_ atoms were obtained through analysis of conventional 3D triple-resonance NMR spectra, while amino acid side chains were assigned using 3D ^15^N- and ^13^C-resolved nuclear Overhauser enhancement (NOE) spectra. This procedure allowed obtaining virtually complete chemical shift assignments (99.8% for the backbone nuclei H_N_, ^15^N, ^13^C_α_, H_α_ and 98.9% for non-labile sidechain hydrogens) (see Appendix A for assigned 2D ^15^N-^1^H heteronuclear single-quantum correlation (HSQC) spectrum). Analysis of ^13^C_α,β_ chemical shift deviations from random coil values showed that 29 N-terminal residues of BBA03 following the signal peptide are disordered (Figure 1a). To obtain higher NMR spectral quality, the unstructured region was omitted from the expression plasmid, and the truncated protein BBA03_53–169_ was used for structure determination. Structure calculation based on NOE distance constraints then resulted in a well-defined structure (structural statistics are given in Table 1).

### 2.2. Structure Analysis

The NMR structure shows that the folded part of BBA03 comprises 115 residues and forms a seven-helix bundle (Figure 1b,c). Analysis of the structure using CASTp software [12] resulted in identification of two clefts with solvent accessible volume larger than 20 Å^3^ (Figure 1c–e). Mole 2.5 [11] online server found that clefts 1 and 2 form entrance to a 17.7 Å long T-shaped hydrophobic pore (Figure 1c). Both clefts are positively charged. The hydrophobic nature of the pore and the positive charge at its entrances suggests that it may bind lipids.

To investigate the possible binding of lipids to the hydrophobic pore of BBA03, we performed molecular docking experiments (Figure 2a,b). Overall, presence of double bonds and increased fatty acid chain length positively affected the ligand binding efficiency calculated as docking score divided by heavy atom count. Four fatty acids (arachidonic acid, arachidic acid, linoleic acid, and oleic acid) showed notably higher ligand efficiency scores as compared to others and formed polar contacts with either R60 or N76 side chains. 

To test if BBA03 binds fatty acids, a 2D ^1^H-^15^N HSQC NMR spectrum was acquired after addition of 10 mM arachidonic acid, palmitic acid, linoleic acid, oleic acid, pelargonic acid, and steric acid to 1 mM BBA03. However, the resulting spectra showed no chemical shift perturbations that would prove the binding of these molecules (Appendix A).

### 2.3. Structure Comparison

A search of structurally similar proteins against Protein Data Bank (PDB) using Dali server [15] (Table 2) revealed that the closest structural homologue to BBA03 is *Borrelia turicatae* BTA121 (Figure 3a). Strong similarity was observed with both polypeptide chains of BTA121 yielding Z-scores of 7.7 and 7.4, and root-mean-square-deviations (RMSDs) of 2.3 Å and 2.6 Å, respectively. BTA121 is encoded on the 150 kDa mega plasmid and, unlike BBA03, its expression is increased in the tick vector. Experiments in vitro have shown that BTA121 binds palmitic acid, but the binding site is unknown [16]. Analysis of BTA121 structure using Mole 2.5 [11] online tool found a similar T-shaped pore as in BBA03 (Figure 3b). 

A high structural similarity was observed also with BBP28, a *B. burgdorferi* mlp family outer surface lipoprotein (Figure 3c) [17]. The main difference between the structures of BBA03 and BBP28 is the lack of the third α-helix in BBP28 and the replacement of the last α-helix in BBA03 with a highly variable extended loop in BBP28. Even though the structural similarity of BBA03 and BBP28 is lower than BTA121, the sequence similarity is substantially higher (see Table 2).

Significant structural similarity is also observed with several Pfam54_60 proteins [18]: BBA64, BBA65, BBA66, BBA69, BGA71, BBA73, and CspA, all of which are α-helical (Figure 3d). Conserved amino acids among them are mostly hydrophobic and are located in the hydrophobic core of the protein, indicative of their function in structural conservation [18]. Thus, despite having a similar fold, Pfam54_60 members have diverse functions. For example, the fifth most similar protein to BBA03, CspA, binds complement factor H [19], while another Pfam54_60 member, BBA70, binds plasminogen [20].

### 2.4. Sequence Comparison

The sequence of BBA03 is well conserved among different genus *Borrelia*. Sequence alignment with the five most structurally similar proteins identified in Table 2 shows that more than half of the conserved amino acids are hydrophobic. Additionally, several positively charged amino acids (R60, K68, K164) are highly conserved, although the positive charge on amino acids 60 and 164 is conserved only in one chain of BTA121. Interestingly, although not all of the compared proteins contain a T-shaped pore, the well conserved R60 residue is located in the positively charged cleft near the entrance to the pore of BBA03. Overall, the highest conserved regions are found in helices α1-α4 as well as the N-terminal parts of α5 and α6 (Figure 4).

## 3. Discussion

*B. burgdorferi* outer surface lipoproteins are the most easily accessible antigenic proteins to mammal immune system during infection. For this reason, they are the primary targets for development of vaccines against Lyme disease. In this work, the solution structure of *B. burgdorferi* outer surface lipoprotein BBA03 was determined revealing a fold with seven α-helices most similar to that of *Borrelia turicatae* BTA121, which has been shown to bind palmitic acid. Remarkably, both protein structures feature two clefts on their surface, which are connected through a long T-shaped hydrophobic pore that could possibly bind lipids. Docking of fatty acids into BBA03 showed that presence of double bonds and increased fatty acid chain length improves ligand binding efficiency. Four fatty acids with the highest binding efficiency were found to dock in the region of the hydrophobic pore, forming hydrophobic contacts as well as polar contacts between the carboxylic group and the surface exposed residues. NMR titration experiments showed no interaction between BBA03 and linoleic acid, oleic acid, arachidonic acid, pelargonic acid, palmitic acid, or steric acid. This does not exclude the possibility of another fatty acid or hydrophobic molecule binding to BBA03.

Another class of structurally similar proteins to BBA03 are the members of Pfam54_60, which have gained their name from their location on the lp54 plasmid [9,19,21,22,23,24,25,27,28,30]. Despite having high structural similarity, the sequence similarity among Pfam54_60 members is rather low, averaging to around 18%, which has resulted in high functional diversity [18]. The mlp family member BBP28 also has a high structural similarity to BBA03, but it is located on ϕBB-1 prophage cp32-1 plasmid [32,33]. The overall fold conservation among proteins without functional relevance is a result of major genetic variations and recombinations in the *Borrelia* genome, which has produced a large number of paralogue gene families [5]. One such variation is the insertion of the cp32 plasmid into the lp54 precursor, which probably resulted in formation of BBA03. Due to the high structural similarity of functionally divergent lipoproteins in *B. burgdorferi*, determination of function is rarely possible by mere structural comparison. The task is even further complicated by lack of sequence homologues among other bacteria [32,34]. 

To our knowledge, BBA03 is the only protein that plays an important role during tick transmission that is manifested only when *B. burgdorferi* is in a mixed infection with other spirochetes in the tick body [8]. A similar phenotype is displayed by BBA07, BBA64, and BBA66, which are located on lp54 and have an unknown function. *Borrelia* lacking these proteins have shown attenuation of infection by tick vector [8,9,29,35], whereas ability of *Borrelia* to infect mammals through a needle inoculation was not affected, similarly as in the case of BBA03 [36,37]. Furthermore, neither the ΔBBA01–07 nor the ΔBBA64 deletion affected the *Borrelia* transition from tick midgut to salivary glands [8,9,35], indicating that BBA03, BBA07, and BBA64 all perform their respective functions in the period of transition from tick vector to mammal host. Such overlap of phenotype indicates a possibility that these proteins have similar or connected functions. This hypothesis is even furthered by the fact that, due to high genetic recombination rate, the functions of *Borrelia* OSPs often tend to overlap [7,38].

Besides providing a competitive advantage to *Borrelia burgdorferi*, the deletion of BBA03 slows down their colony growth rate, and the colonies grown are more diffuse [8]. As BBA03 is an outer surface lipoprotein, which does not form homodimers, this finding indicates that it either interacts with a compound in the medium or with another molecule on the surface of *Borrelia*. Research investigating BBA03 effect on colony growth used BSKII medium, which contains a fatty acid source such as 6% rabbit serum. Therefore, the hypothesis that BBA03 binds fatty acids would be in consistence with this finding. Proteins with such function are necessary for *B. burgdorgferi,* as they lack the enzymes necessary for cholesterol and other fatty acid biosynthesis or elongation [39,40]. To the authors’ knowledge, there are no known fatty acid binding *B. burgdorferi* lipoproteins. Furthermore, complex fatty acids gained from the environment do play an important role in the formation of complex lipid rafts structures found on the surface of *Borrelia* [41,42]. Previous research in fact shows that *B. burgdorferi* do exchange cholesterol with mammal cells through vesicles and direct contact [43].

In conclusion, we determined the solution NMR structure of BBA03, a *Borrelia burgdorferi* competitive advantage promoting outer surface lipoprotein. On its surface, a T-shaped hydrophobic pore was found, which could bind lipids. Although docking experiments suggested possible binding of long-chain fatty acids in the pore, NMR titration experiments were performed but showed no interaction. Based on our findings, we hypothesize that BBA03 confers competitive advantage to *Borrelia* by enhancing the fitness of the bacteria. Future binding studies with the components of BSKII medium or additional lipid molecules need to be performed in order to determine the exact function of BBA03.

## 4. Materials and Methods

### 4.1. Protein Sample Preparation

BBA03 was produced from pETm-11 plasmid [44], omitting the first 22 residues encoding the signal sequence. Constructs comprising residues 23–169 (with tether region) and 53–169 (only structured region) were used for protein backbone resonance assignment and structure determination, respectively. The second construct containing only the structured part was obtained by amplifying the part excluding the N-terminal tether region and ligating back into the pETm-11 vector using the primers 5′- ataccatgggtacacctttagaaaaattagt -3′ and 5′- cagcctcgagctatatagtgtctttaagtttat -3′, where the NcoI and the XhoI restriction sites were underlined. The obtained plasmid was used in BL21DE3 *E. coli* cell transformation, and the doubly labeled proteins were expressed in M9 minimal media containing 2g/l [^13^C_6_]-d-glucose and 1 g/l ^15^NH_4_Cl (Cambridge Isotope Laboratories) as sole sources of carbon and nitrogen, respectively. The cells were grown at 37 °C in the presence of 50 mg/l kanamycin and induced when absorbance at 600 nm (OD_600_) reached ~0.6. After 4 h of expression, *E. coli* were harvested at 10000 g, 4 °C for 10 min and frozen at −20 °C. The cell pellets were resuspended in 40 mL buffer A (20 mM sodium phosphate, pH 7.5, 300 mM NaCl, 15 mM imidazole) and were lysed by ultrasonication. Soluble cell lysate fraction was separated by centrifugation for 40 min at 25,000× *g*, 4 °C and filtration thorough a 0.22 μm pore-sized filter. The solution was loaded onto a 5 mL HisTrap HP column (GE Helthcare). The bound protein was eluted using imidazole gradient from 15 to 300 mM over 30 mL volume. Fractions containing BBA03 were concentrated to 1 mL using ultra-centrifugal filter units with 10 kDa molecular weight cutoff. To remove N-terminal His-tag, 25 μg/mL of tobacco etch virus (TEV) protease were added, and the solution was kept overnight at 4 °C. Afterwards, it was passed through a HisTrap HP column and concentrated to 550 μL in NMR buffer (20 mM sodium phosphate, pH 6.8, 50 mM NaCl). Each purification step was monitored by SDS-PAGE.

NMR samples were prepared by adding 5% (*v*/*v*) D_2_O and 0.03% NaN_3_ to a 1.5 to 2 mM ^13^C,^15^N-labeled BBA03 solution. For lipid binding experiments, 10 mM fatty acids were added to the NMR sample tube containing 1 mM protein and mixed thoroughly before performing NMR measurements. Palmitic acid was dissolved in 30 μL of ethanol before addition to the sample (final concentration of ethanol in the sample was 5%).

### 4.2. NMR Spectroscopy

NMR experiments were conducted at 298 K on a Varian Unity Inova 600 MHz spectrometer equipped with a TCI HCN z-gradient cryoprobe or Bruker Avance III HD 800 MHz spectrometer equipped with an TXI z-gradient room temperature probe. For backbone assignment, 3D HNCA, 3D HNCO, 3D CBCA(CO)HN, and 3D HN(CA)CO spectra were acquired. 3D [^1^H,^1^H]-NOESY-^15^N-HSQC, 3D [^1^H,^1^H]-NOESY-^13^C(aliphatic)-HSQC, 3D [^1^H,^1^H]-NOESY-^13^C(aromatic)-HSQC spectra were acquired using a mixing time of 80 ms. Chemical shifts were referenced internally to the residual water signal at 4.77 ppm relative to 4,4-dimethyl-4-silapentane-1-sulfonic acid (DSS).

### 4.3. NMR Relaxation Data

Residue-specific protein backbone amide ^15^N NMR R_1_ and R_2_ relaxation times were calculated from peak intensities in ^15^N-T_1_ and ^15^N-T_2_ (gNT1T2) relaxation spectra using two-parameter exponential fit model in relax 4.0.3 [45]. Nine T_1_ (10 to 1500 ms) and seven T_2_ (10 to 210 ms) delays were used for the recording of the decay curves. Protein rotational correlation time (τ_c_) was calculated using the following equation: $\tau_{c}\approx \frac{1}{4{\;\pi v}_{N}}\sqrt{6\frac{T_{1}}{T_{2}}-7}$ [46], where ν_N_ is the ^15^N resonance frequency (in Hz), T_1_ is average longitudinal, and T_2_ is average transverse relaxation time value of rigid protein regions. 

### 4.4. NMR Structure Determination

Spectra were processed using TopSpin 3.5pl7 software. Chemical shifts of protein backbone and side chain nuclei were assigned by analysis of all the acquired spectra in CARA [47]. NOE upper distance limits were automatically obtained from UNIO ATNOS/CANDID 2.0.2 [48,49,50], which was used together with NMR structure calculation algorithm CYANA 2.1 [51]. Seven cycles of UNIO ATNOS/CANDID protocol were used, starting with 100 random conformers that were subjected to simulated-annealing with 10,000 steps of torsion-angle molecular dynamics. The 25 conformers with the lowest residual CYANA target-function values after UNIO ATNOS/CANDID cycle 7 were energy minimized in explicit water using CNS [52]. The coordinates of the final 20 NMR conformers and all restraints were deposited in the Protein Data Bank (accession code 6QS0), and chemical shifts were deposited in BMRB (accession code 34361).

### 4.5. Molecular Docking

The solution NMR structure of *B. burgdorferi* outer surface lipoprotein BBA03 determined in this study was prepared for docking using the protein preparation wizard tool in Maestro 12.4 [53]. Fatty acids (C10 to C20 saturated fatty acids, as well as arachidonic, linoleic, and oleic acids) were prepared for docking using the standard protocol implemented in LigPrep. 

Docking was performed using the Induced Fit Docking tool implemented in Maestro Schrodinger software. Centroid of Leu27, Leu73, and Leu74 was selected as a docking box center. Docking box length was set to 30 Å. Protein and ligand van der Waals radius scaling was set to 0.5. Up to 20 poses per ligand were retained after the initial docking and were further minimized by refining the protein residues within 5.0 Å from the ligand. OPLS3e force field [54] was used for the complex minimization. Docking scores were calculated by re-docking ligands in the top 5 optimized proteins structures at a standard precision using Glide. Ligand efficiency was calculated as docking score per ligand heavy atom count.

## Figures and Tables

**Figure 1 pathogens-09-00826-f001:**
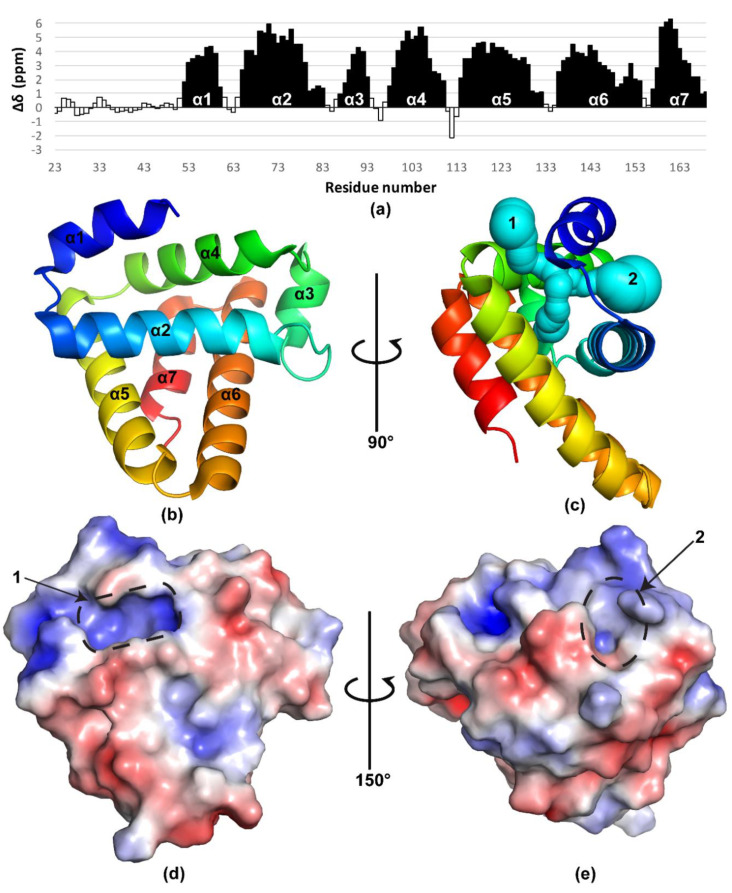
Structure of BBA03. (**a**) Secondary chemical shift analysis of BBA03. Here, Δδ_i_ is calculated as (ΔC_α_ - ΔC_β_) + (ΔC_α − 1_ - ΔC_β − 1_) + (ΔC_α + 1_ - ΔC_β + 1_) / 3. Δδ_i_ values above 1 for three or more consecutive residues are indicative of alpha helix (black) and below −1 of beta sheet secondary structure, whereas Δδ_i_ values between −1 and 1 indicate non-regular structure (white). (**b**,**c**) Cartoon representation of BBA03 structure colored using rainbow color scheme. It starts with blue color at N-terminus and gradually changes to red at C-terminus. In (**c**), the T-shaped pore identified by Mole 2.5 [11] online tool is shown in cyan. Clefts 1 and 2 found by CASTp software [12], larger than 20 Å^3^, are denoted with numbers. The volume of the first cleft is 20.4 Å^3^ and of the second is – 76.8 Å^3^. (**d**,**e**) Electrostatic potential of BBA03 surface, calculated using Adaptive Poisson-Boltzmann Solver (APBS) Pymol plugin [13]. The gradient starts from red, where 5 kBT/e < 4, and moves to deep blue, where 5 kBT/e > 4. Above mentioned clefts are marked with a dashed line.

**Figure 2 pathogens-09-00826-f002:**
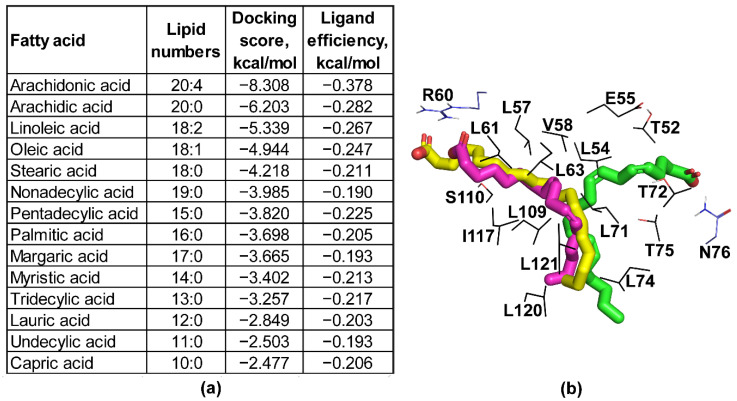
Docking of fatty acids to BBA03. (**a**) List of fatty acids docked to the determined structure of BBA03 and their ligand efficiency. (**b**) Arachidic acid (yellow), arachidonic (green), and linoleic acid (magenta) docked to BBA03 (dark grey). Pore forming amino acid side chains (T52, L54, E55, L57, V58, R60, L61, L63, L71, T72, L74, T75, N76, L109, S110, I117, L120, and L121) are shown with thin lines and labeled. Side chains which form polar interactions with the fatty acids (R60 and N76) are colored blue, while other pore forming amino acids are colored in black.

**Figure 3 pathogens-09-00826-f003:**
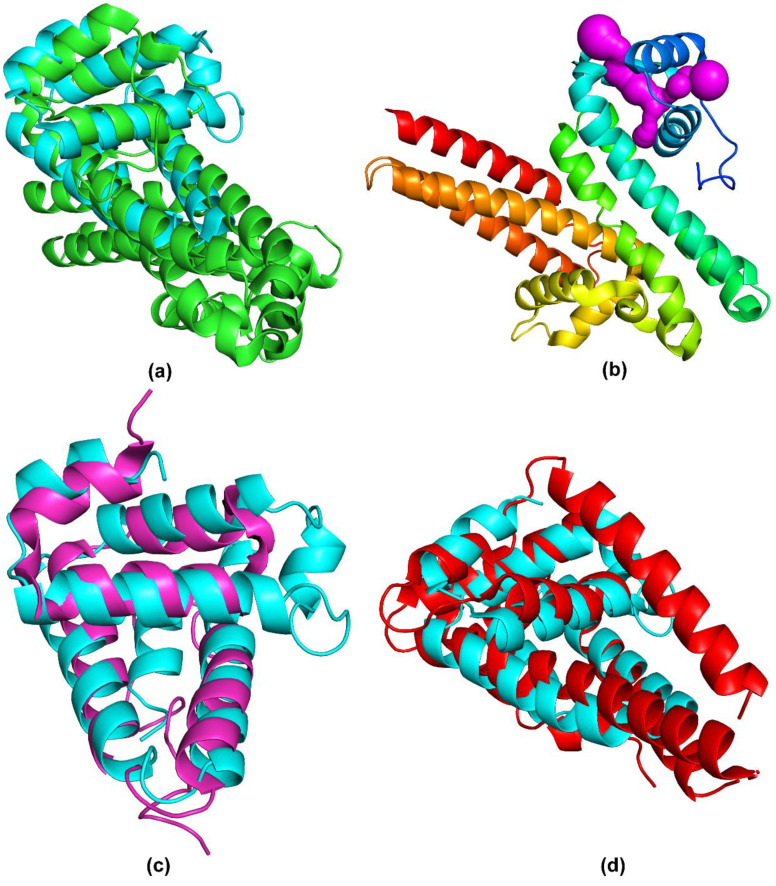
Comparison of BBA03 with structurally similar proteins found by Dali server. Superposed structures of BBA03 (cyan) and BTA121 (PDB: 5VJ4) (**a**,**b**), BBP28 (PDB: 6QBI) (**c**), BGA71 (PDB: 6FL0) (**d**). In (**b**) BTA121 is colored using rainbow color scheme. T-shaped pore in BTA121 identified by Mole 2.5 [11] online tool is shown in magenta.

**Figure 4 pathogens-09-00826-f004:**
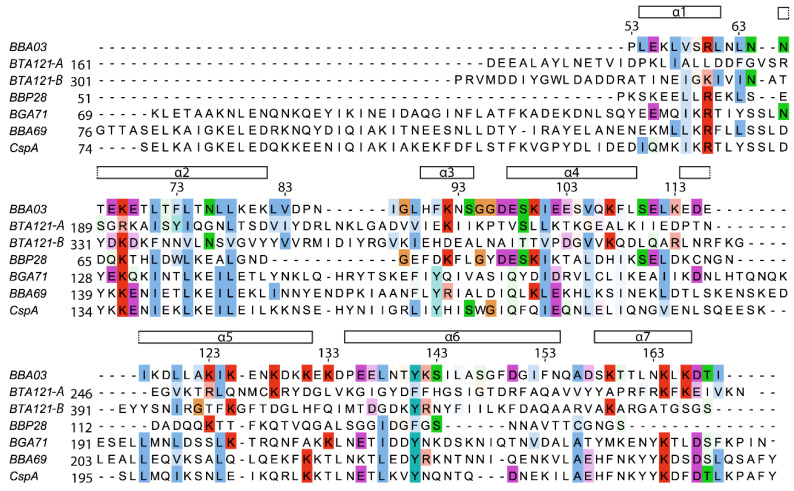
Sequence comparison of BBA03 with structured parts of five structurally most similar proteins from *Borrelia* genus: BTA121 (GenBank: YP_009164930.1), BBP28 (GenBank: WP_010883731.1), BGA71 (GenBank: WP_011187214.1), BBA69 (AAC66287), and CspA (AAC66286). [31]. Both polypeptide chains of BTA121 (A and B) are shown in the alignment. BBA03 secondary structure elements and residue numbering are indicated above the alignment. Residues that are conserved or the same type as in BBA03 are highlighted in blue (A, I, L, M, F, W and V), turquoise (Y and H), green (N, Q, S and T), red (R and K), purple (D and E), orange (G), pink (C) and yellow (P). Lighter coloring indicates lower similarity.

**Table 1 pathogens-09-00826-t001:** Input data for BBA03_53–169_ structure calculation and structure quality statistics.

NOE upper distance limits	2602
Intra-residual (|i − j| = 0)	767
Sequential (|i − j| = 1)	626
Medium range (1 < |i − j| ≤ 4)	704
Long range (|i − j| ≥ 5)	616
Residual CYANA target function value after CNS (Å^2^)	9.75 ± 1.05
**Residual NOE violations**	
Number ≥ 0.1 Å	29 ± 5
Maximum (Å)	0.38 ± 0.08
**PARALLHDG force field energies (kcal/mol)**	
Total	−5534 ± 101
Van der Waals	−481 ± 19
Electrostatic	−5961 ± 117
**Root-mean-square-deviation (RMSD) from mean coordinates (residues 53–169) (Å)**	
Backbone	0.50 ± 0.06
All heavy atoms	0.90 ± 0.06
**Molprobity structure statistics** [14]	
Ramachandran plot favored regions (%)	94.5
Ramachandran plot allowed regions (%)	99.3
Average clash score of all atoms and models	21.62 ± 2.47

NOE: nuclear Overhauser enhancement.

**Table 2 pathogens-09-00826-t002:** Structural similarity of BBA03 to other proteins from *Borrelia* genus bacteria.

Protein	PDB Code	Z-score	Backbone RMSD, Å	Sequence Identity %	Function
BTA121 [16]	5VJ4	7.7	2.3	13	Binds fatty acids [16]
BBP28 [17]	6QBI	7.4	2.5	21	Unknown
BGA71 [21]	6FL0	7.0	2.9	12	Binds C7, C8 and C9 terminal complement components [19]
BBA69 [22]	6QO1	6.1	3.2	14	Unknown
CspA [23]	5A2U	6.1	3.3	11	Binds factor H and FHL-1 [24] Binds C7 and C9 terminal complement components [25]
BBE31 [26]	6FZE	5.4	3.1	9	Binds glutathione [26]
BBA64 [23]	4ALY	4.3	3.4	18	Unknown [9]
BBA65 [18]	4BG5	4.3	3.4	14	Unknown [27]
BBA66 [28]	2YN7	4.3	3.6	12	Unknown [29]
BBA73 [30]	4B2F	4.1	3.5	13	Unknown

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
