# Peer review of "Structural and Functional Analysis of BBA03, Borrelia burgdorferi Competitive Advantage Promoting Outer Surface Lipoprotein"

_pathogens, 2020, doi:10.3390/pathogens9100826_

Round 1

Reviewer 1 Report

The review in the file below.

Reviewer 2 Report

A previous work from Bestor et al reported that BBA03, a putative outer surface protein encoded in lp54 plasmid, provides Borrelia burgdorferi a competitive advantage to complete its transmission from the tick vector to a mammalian host; however, the underlying mechanism remains unknown. In this report, Fridmanis et al solved the solution structure of BBA03 using NMR. The obtained structural data suggests that BBA03 is a pore forming protein that may bind lipids, which provides insights into understanding the role of BBA03 in the pathophysiology of B. burgdorferi. Overall, the structural part (NMR, docking, and homology modeling) is solid, but its scientific merit has been limited by lacking of functional study. It needs more experimental data to justify the conclusion. Specifically, while docking analysis suggests BBA03 binds fatty acids, the NMR analysis showed otherwise. Also, it remains questionable if the identified two clefts can form a pore.

Minor comments

  1. Figure 1, it would be more informative if the amino acids involved pore formation and ligand (fatty acids) binding were highlighted. In addition, an electrostatic surface model can be included to highlight those positive charged residues. 
  2. Figure 2, in addition to R60 and N76, are there any other residues involved the binding?
  3. Figure 3, PDB ID numbers for these three superimposed proteins should be provided in the figure legend.
  4. Figure 4, GenBank accession numbers for those aligned proteins should be included in the figure legend.
  5. Line 177-178, “several positive charged amino acids (R60, K68 and K164) are highly conserved”. This is debatable as some of these residues are variable. For example, R60 is missing in BTA121-A and BTA121-B albeit they have the highest structural similarity to BBA03 (smallest RMSD) among those 10 aligned proteins. 

Editorial suggestions:

Line 28, to be specific, replace “vector” with “tick” or “arthropod”

Line 39, “protease K” should be “proteinase K”.

Line 44, change “transmission” to “migration”.

Round 2

Reviewer 2 Report

Previous concerns have been adequately addressed. There is no further concern.